# Assessing Driving Risk Using Internet of Vehicles Data: An Analysis Based on Generalized Linear Models

**DOI:** 10.3390/s20092712

**Published:** 2020-05-09

**Authors:** Shuai Sun, Jun Bi, Montserrat Guillen, Ana M. Pérez-Marín

**Affiliations:** 1Key Laboratory of Transport Industry of Big Data Application Technologies for Comprehensive Transport, School of Traffic and Transportation, Beijing Jiaotong University, Beijing 100044, China; bilinghc@163.com; 2Department of Econometrics, Riskcenter-IREA, Universitat de Barcelona, 08034 Barcelona, Spain; mguillen@ub.edu (M.G.); amperez@ub.edu (A.M.P.-M.)

**Keywords:** internet of vehicles, usage-based insurance (UBI), driving risk, driver classification, regression analysis

## Abstract

With the major advances made in internet of vehicles (IoV) technology in recent years, usage-based insurance (UBI) products have emerged to meet market needs. Such products, however, critically depend on driving risk identification and driver classification. Here, ordinary least square and binary logistic regressions are used to calculate a driving risk score on short-term IoV data without accidents and claims. Specifically, the regression results reveal a positive relationship between driving speed, braking times, revolutions per minute and the position of the accelerator pedal. Different classes of risk drivers can thus be identified. This study stresses both the importance and feasibility of using sensor data for driving risk analysis and discusses the implications for traffic safety and motor insurance.

## 1. Introduction

With the advances made in information technology, the internet of things (IoT) is gradually revolutionizing the way people live. An important part of this advanced network is the internet of vehicles (IoV) which, based on telecommunications and informatics technology, collects data about vehicles, roads, and the environment. These data, collected from multiple sources, are then processed and shared. For example, data can be analysed by insurance providers to improve products and services. In short, the IoV facilitates information sharing through vehicle-to-vehicle (V2V), vehicle-to-person (V2P) and vehicle-to-road (V2R) interconnectivity [1].

Our objective here is to show that IoV data can be an important source of driving risk indicators, which can be used by drivers as inputs for improving their driving habits and by insurers for calculating insurance premiums. We also propose that risk indicators can be incorporated into the internal architecture of a vehicle’s sensors. We do not consider autonomous vehicles. Ryan et al. address the quantification of risk for autonomous vehicles compared to humans [2]

IoV data are typically collected by means of on-board devices: pre- and post-installed devices and smart phones being the three of the most common data acquisition terminals. A pre-installed device is the electronic component that an automobile is equipped with when it leaves the factory; a post-installed device is added by users according to their specific business demands once the vehicle has left the factory and a smart phone is carried by a driver and uses a satellite-positioning module, acceleration sensor, magnetic sensor, proximity sensor, gyroscope and other sensors to collect data when driving. All three sources of data collection have their respective strengths and weaknesses, but they can all transmit the data to the cloud via remote communication technology. 

The big data provided by the IoV are opening up new opportunities to accelerate innovation in the automobile insurance business. For example, usage-based-insurance (UBI), the personalized insurance product based on IoV data, calculates insurance premiums according to drivers’ driving patterns rather than using questionnaires collecting a priori basic vehicle and personal information. High-risk drivers pay a higher insurance premium, while more cautious, low-risk drivers obtain the corresponding insurance premium privileges. 

Based on the criteria used in assessing driving risk and the timeliness of IoV data, UBI can be divided into different categories. Pay-as-you-drive (PAYD), the first UBI product to be launched to market, assesses driving risk based solely on mileage and fuel consumption, and charges a premium proportional to that risk. Subsequently, given the large amounts of driving behavior data obtained from the IoV, pay-how-you-drive (PHYD) insurance products have been developed and marketed. Compared to PAYD, PHYD takes into consideration more indicators of a driver’s behavior, including acceleration, braking, cornering and speed, which allows driving risk to be assessed more accurately and premium discounts to be calculated more precisely. As information technology and the IoV have become more elaborate, determining driving risks and calculating real-time insurance premiums while collecting data have become possible. In manage-how-you-drive (MHYD), data-processing results and occasional vehicle failure information can be fed back to drivers. Compared to PHYD, MHYD offers better timeliness and functionality in data processing and application, but is more complex.

This study focuses specifically on how PHYD applications can be exploited under various conditions. Given that the principal idea underpinning PHYD plans is to offer premium discounts to low-risk drivers, driving risk should be accurately and effectively assessed. To do so, risky drivers have to be identified, a task that is greatly facilitated by scoring all drivers on a one-dimensional scale, where a high score identifies a high-risk driver and a low-score identifies a low-risk driver. 

To date, machine learning (ML) methods, including decision tree, random forest, support vector machine, neural network regressions and generalized linear models (GLMs), among others, have been applied to studies of PHYD plans [3,4,5,6]. However, as most ML algorithms are black-box algorithms, and insurance prices are highly regulated around the world, some ML methods have not gained much popularity to date. Essentially, black-box procedures cannot be expressed in a predictive parametric formulation so that insurance premiums can be calculated based on an analytic transformation of a weighted combination of factors. GLMs are more familiar to insurance practitioners and customers as regards the practical application of PHYD products. Moreover, GLMs capture the parameterized relationship between driving risk and related factors, which is especially useful for further research into driving risk identification and the communication of features leading to low-risk driving patterns.

In contrast with most previous studies [4,6], the driving behavior data used here to identify risky drivers are short-term and do not include accidents and claims. This undoubtedly increases the difficulty of the study, as we have no label data and so we must customize an indicator that can represent driving risk. However, having said that, in the real-world application of UBI, the user data obtained by an insurance company are likely to be short-term and contain no information about accidents and claims because of limited acquisition time and privacy issues. 

As such, our study is of interest to insurance companies, as they typically have to use imperfect driving behavior data to identify driving risk and calculate fair premiums. Moreover, our study highlights the importance of sensors in the IoV.

The rest of this article is organized as follows. Existing research, conducted both in academia and in the insurance industry, examining the application of UBI are summarized in Section 2. Data sources, data forms, and data preprocessing are described in Section 3. The conventional multiple linear regression model and logistic regression model used for scoring driver risk are described in Section 4. The classification results obtained from different algorithms are reported and compared in Section 5. These results are discussed in Section 6 and the conclusions we draw are presented in Section 7.

## 2. Literature Review

Generalized linear models are typically used to study relationships between variables and a response. For example, Jin et al. built two binary logistic models to examine the effects of telematics-related variables on predicting the probability of traffic accidents [7]; Verbelen et al. used generalized additive models and compositional predictors to study the effect of different telematics variables on the expected claim frequency [3]; and Ma et al. used Poisson regression to analyze the relationship between several risk factors and traffic accident probability [8]. Conventional generalized linear regression has always played an important role in the study of driving risk because of the interpretability of its variables. Note, however, that in many countries, premium calculation is regulated and, as a result, the authorities prefer non-black-box methods [9].

With the rise of machine learning, various studies have employed different ML methods to identify driving risks. For example, Carfora et al. assessed driver aggressiveness on urban and highway roads using cluster analysis [10] and Jafarnejad et al. showed that AdaBoost classifiers can provide good predictions of driver identification based on telematics data [11]. Paefgen et al. compared the performance of various ML methods, including logistic regression, neural network, and decision tree classifiers, for driving risk prediction and insurance pricing, and found that logistic regression models had advantages in terms of both accuracy and interpretability [12]. Finally, Bian et al. employed the ensemble learning method to build a driver risk classification model [13]. In most ML studies, logistic regression is selected as the method for driver scoring or driving risk classification and it has been reported to perform well.

The number of indicators of driving risk is increasing with the development of sensor technology. Traditional automobile insurance often relies on quite basic information about the customer and the vehicle. In some advanced markets, prices usually include experience rating systems, which reward good drivers and vehicle type or regional classes; however, with the development of UBI, PAYD products require the collection of mileage and fuel consumption data, as recorded on the odometer, with some studies indicating that the greater the mileage, the higher the probability of suffering an accident [14]. PHYD products have also begun to take into account driving behavior data (including speeding, harsh acceleration, sudden braking and sharp turns) and driving condition data (including road types and night-time driving), which have been shown to be associated with driving risk; however, as yet, there are not many mature PHYD products on the market [4,15,16].

In practice, the use of IoV data is not without its problems. While on-board devices can collect data indicative of many key features, they are not universally accepted due to privacy or installation issues [17,18]. Moreover, different on-board devices have different characteristics [16], and while many UBI studies have been unable to obtain all these key features, driving risks and insurance pricing can still be assessed with these limited characteristics [3,19]. Smart phones, on the other hand, while offering advantages of portability, high efficiency and flexibility, have seen their development checked by a lack of data integrity compared to that offered by other on-board devices. Additionally, smartphone data are often missing braking and fuel consumption measurements; even so, UBI can still be attempted [20]. Although the data used in this study do not include accidents and claims, they do contain what can be considered key driving behavior data. Many studies report a positive correlation between accidents and excessive acceleration and braking and so, here, we use these data to undertake our analysis of driving risk [21].

Zuo et al. developed a vehicle terminal that can accurately obtain the vehicle’s behavior information, including acceleration, deceleration, sharp turns, and sudden lane change [22]. They suggested that statistical analyses of these data produce a broad application prospect in the field of UBI. Recently, Li et al. proposed a method to prove that physiological and vehicle information (average speed) have a significant influence on the driving risk [23]. However, better pricing based only on driving behavior is only accessible if internal data of the vehicle are combined with map/street and weather data, and it might be necessary to have also additional data from other sources (other vehicles, traffic jam cameras, speeding limit in areas of road work, etc.).

## 3. Data Description

The IoV data used in our study were obtained from an IoV information service provider in China. The IoV information service provider is called ‘China Satellite Navigation and Communications Co., Ltd.’ which is a high-tech enterprise providing intelligent network technology services, headquartered in Beijing. The IoV information service gathers data from their own intelligent network vehicles. Data acquisition, transmission, storage, and application procedures are shown in Figure 1. Each vehicle was pre-installed with a telematics box (T-box), which included a GPS sensor, vehicle condition sensor, and wireless transmission unit. When the vehicle was started up, information was updated second-by-second and aggregated at the device level to reduce data transmission and storage costs. Every 30 s, the T-box transmitted the last data item to the database. When the vehicle was shut down, the on-board device automatically restarted every 30 min and transmitted a data item to the database. The data from the database include unique vehicle identification, time-varying GPS trajectory data, and vehicle condition information (for example, mileage, fuel consumption, speed, accelerator pedal position, brake times and so on); they do not include accidents and claims information, which are unavailable for privacy and legal reasons. However, based on the assumption that some indicators are associated with the frequency of accidents, the information can be used to score drivers and to assess accident risks.

Figure 2 shows the operation of a vehicle from 0:00 h on 4 July 2018 to 0:00 h on 5 July 2018. The vehicle’s recorded mileage increased from 22,812.35 to 23,317.33 km, and its accumulated consumption increased from 8589.69 to 8779.67 L. At the lower part of Figure 2, we display the speed. We see that the car is idle during the night. During this period, driving speeds varied from 0 to 110 km per hour. Acceleration and braking actions occur at different speeds. Figure 3 presents an example of the trajectory associated with a vehicle in the course of one day. 

Our dataset comprises 309 data files for 309 unique vehicles over a five-day period. These files were obtained in two groups according to the time span: that of the first group runs from 0:00 h on 27 June 2018 to 10:00 h on 2 July 2018, while that of the second group runs from 0:00 h on 3 July 2018 to 22:00 h on 8 July 2018. Since the timespan of both groups is about five days and these measurement periods correspond to two adjacent weeks, it is reasonable to combine the two groups into one. We assume that weather and other relevant conditions are almost identical in these two periods for the two groups. Some data files are empty due to transmission errors or for reasons unknown. Therefore, the total number of effective vehicle data files after excluding the empty data files is 253.

Each data file contains 62 variables, but not all the latter can be used directly. First, some variables have many null or meaningless invariant values, due to either sensor failure or extraction error. Second, some variables are redundant. Indeed, some of the variables are redefined and calculated by the information provider, as transformations of the original variable. In addition, some variables have outliers. After further data processing, the following key variables were selected for this study (see Table 1).

## 4. Methods

### 4.1. Determination of Dependent Variables

As discussed, our data do not contain any accident or claims information that can be used as dependent variables or labels. However, with the exception of a few unsupervised ML learning algorithms (such as clustering algorithms) that do not require label data, both GLMs and supervised machine learning algorithms require a dependent variable or label data, otherwise they cannot be implemented. Therefore, it is imperative at the outset to define a dependent variable to represent driving risk. 

Excessive braking while driving, especially at high speeds, indicates that the vehicle is likely to be exceeding the safe speed limit. When driving at an exorbitant speed and foreseeing danger, a driver is obliged to brake, as opposed to releasing the accelerator pedal, to avoid a collision. Thus, to some extent, braking can be considered a high-risk driving behavior and, so, for the purposes of this study, a driver’s brake count was selected as the dependent variable.

Acceleration is an essential part of driving, but different drivers have different habits as regards acceleration. Drivers with good driving habits tend to minimize unnecessary harsh acceleration, thus controlling the vehicle’s speed more effectively and saving fuel in the process. Poor drivers, on the other hand, do not have the same control over the use of the accelerator, resulting in more braking to curb their speed. Excessive harsh acceleration increases driving risk. Therefore, the average position of the accelerator pedal (the higher the value, the greater the frequency of harsh accelerations) was chosen as a second dependent variable.

In the logistic regression model, the dependent variables, both “Brakes” and “Accelerator”, were dichotomized as follows. Value levels above the median were identified as events equal to one, while those below the median were identified as non-events equal to zero.

### 4.2. Determination of Independent Variables

As discussed, we were left with a number of effective variables after data preprocessing. Among these, driving distance, speed and RPM were selected as the independent variables for our driving risk evaluation. Driving distance and fuel consumption are two of the earliest factors used to study PAYD. Longer distances and higher fuel consumption mean a higher risk of accidents. Because of the strong correlation between driving distance and fuel consumption, we selected driving distance as an independent variable. In addition, with the exception of harsh acceleration and sudden braking, speed is often selected as a driving risk factor: the higher the speed, the harsher the acceleration a driver needs, and the more braking he needs to bring the vehicle to a stop [24]. In this study, the average speed of each vehicle is used as one of the independent variables. Finally, RPM, an infrequently employed variable in these studies, reflects, to some extent, driving intensity, and so the average RPM of each car is selected as our third independent variable.

Additionally, we defined a new measure (Range) of driving patterns in this instance, to capture the fact that a driver always remains in the same region (see Figure 4). It was calculated as follows, based on the available GPS trajectory
(1)$$Range=\sqrt{{\left(long_{max}-long_{min}\right)}^{2}+{\left(lat_{max}lat_{min}\right)}^{2}}$$
where *long_max_* and *long_min_* represent the maximum and minimum observed Longitude values, respectively, and *lat_max_* and *lat_min_* represent the maximum and minimum observed Latitude values, respectively.

Theoretically speaking, if drivers brake or accelerate a lot because of traffic congestion or because of having to move in a limited area, this is not indicative of high-risk driving situations. However, if the driver drives over a wide area per unit of time (that is, at high speeds), a high number of brakes or accelerations may represent a safety hazard.

The summary statistics of our original summary data files are presented in Table 2. The important thing to bear in mind is that “Accelerator” (“Brakes”) inputs the regression model as an independent variable if “Brakes” (“Accelerator”) is the dependent variable. Note that, for example, RPM average is low because we also consider the time when the car is idle.

### 4.3. Feature Creation and Extraction

Polynomial variation can be used to increase the variable dimension of data and transform part of the data from nonlinear to linear, thus allowing linear regression to process nonlinear data. However, excessively high dimensions hinder subsequent analyses and increase collinearity among variables. For this reason, the dimension is limited to two in this study. Two-by-two plots are displayed in Figure 5 and Figure 6.

In order to facilitate interpretation of the data analysis results, ordinary least square (OLS) regression and logistic regression were conducted. However, even when controlling for the dimensions of the polynomial variation, the increased number of features still hampered regression analysis. Hence, the stepwise regression was conducted by bidirectional elimination. In this way, we eventually obtained the combination of independent variables that best explains the dependent variables.

## 5. Results

### 5.1. Brakes as Dependent Variable

First, OLS and logistic regressions were carried out by taking “Brakes” as the dependent variable and “Accelerator” as one of the independent variables. The stepwise regression results after bidirectional elimination are shown in Table 3. The adjusted-R^2^ statistic of the OLS regression (0.3420) and the pseudo-R^2^ statistic of the logistic regression (0.2472) are both small.

### 5.2. Accelerator as Dependent Variable

Second, “Accelerator” was selected as the dependent variable, and “Brakes” was selected as one of the independent variables. Then, OLS and logistic regressions were again carried out. The regression results are shown in Table 4. The adjusted-R^2^ statistic of the OLS regression (0.7070) and the pseudo-R^2^ statistic of the logistic regression (0.5321) are greatly improved, indicating that the model offers a more accurate reflection of the relationship between variables. Thus, the best choice of dependent variable for the regressions in this study is “Accelerator”.

## 6. Discussion

The regression model allows us to score and classify drivers based on our input data. The independent variables were substituted into the regression model to obtain the estimates. Taking these results as the ordinate value and the observed dependent variable as the abscissa value, the median of the predicted scores and the median of the observed values are used as classifiers (see the scatter diagram in Figure 7). The x-axis presents the intuitive judgment of the original dependent variable, where zero is “Good”, the median of the observed data is “Median”, and the maximum is “Bad”. Likewise, the y-axis presents the predicted results, where zero is “Good”, the median of the predictions is “Median”, and the maximum is “Bad”. In general, the smaller the value of “Accelerator”, the more cautious the driving style and the better the driver. In most cases, the predicted value corresponds closely to the expected value, or lies close to the observed value. However, a few predictions clearly do not meet expectations.

The points in the bottom left and top right of the scatter diagrams are as expected. Drivers with low “Accelerator” values, whose expected driving risk is also small, are unquestionably good drivers. In contrast, drivers with high “Accelerator” values, whose expected driving risk is correspondingly high, are undeniably risky drivers. Similarly, drivers in the top left are high-risk drivers and, while they did not drive aggressively for a short period, they present a tendency to drive aggressively based on their driving behavior. In addition, the driver in the bottom right also needs special attention. For example, a driver who has an average accelerator pedal position of 30%, but whose risk model predicts 20% or lower, should be warned, because his level of acceleration activity is above the risk level predicted by the model. Therefore, those in the bottom right are also identified by our models as risky drivers.

Table 5 shows the classification results of the two models used in this study. The results of the two classifications are generally consistent, but there are a few inconsistencies that can affect our assessment of risky drivers. Thus, to improve fault tolerance in their predictions, it seems both models should be used, so as to take into consideration their respective high-risk judgments. 

A section of the data should be test out-of-sample to show how the model predicts new data. We have split the sample in a train dataset (75% instances) and a test dataset (25% instances). The training and test samples were chosen randomly. The comparison of predictive results and classification is shown in Figure 8 and Table 6. The conclusions do not change.

Insurance companies seek to charge a premium according to the normal standard calculation of the driving risk of the driver. In the case of low-risk drivers, insurance companies need to offer certain premium discounts to encourage drivers to maintain their current driving habits. However, in the case of high-risk drivers, insurers would be better off charging them punitive premiums and giving them feedback to help them drive more safely.

Our work has some limitations. As suggested by one reviewer, some additional discussion around the speed of the vehicle at the time is important. For example, a harsh braking event can occur when the vehicle is travelling at 5 km/h with no safety implications, while the same harsh braking at 50 km/h is obviously a different proposition. We need to analyze the detailed data instead of averages. Various outliers that clearly do not obey the model can be identified. Explanations for these outliers might include problems with vehicle sensors or with the data being collected, transmitted, or stored. However, giving feedback on these outliers to the information providers should enable them to identify the source of the problem and improve the quality of their sensors and information services.

## 7. Conclusions

This study has demonstrated that IoV data without accidents and claims can be used for undertaking driving risk analyses and driver ratings. As mentioned in the literature review, accidents and the indicators of driving risk are strongly related [14,15,16]. Here, we use variables that are theoretically related to driving risk to replace these dependent variables in a regression analysis and find that when the mean of the accelerator pedal position is used as the dependent variable, the regression model captures the relationship between variables well. This means that insurance companies can use limited data resources for regression analysis when claims data for vehicles are unavailable, but premiums need to be priced.

Using regression analysis, we identified the relationship between our independent and dependent variables that indirectly reflects the influence of various factors on driving risk. Speed is the main factor influencing driving risk: the higher the speed, the more the driver needs to accelerate, the higher the number of RPM, and the more the driver needs to brake, the greater the driving risk. At high speeds, braking action is significantly reduced to avoid sideslip and rollaway accidents caused by braking. However, the range variable specifically created in this study does not appear to play a relevant role. Moreover, the mileage variable performs differently in the two regression analyses, but it does at least show that this variable is related to braking and acceleration, which, in turn, are related to driving risk.

To analyze telematics data and accident data, a comprehensive view of risky driving is needed, Insurers tend to have a complete vision of accidents and, in particular, accident severity combined with the driver’s past experience. Telematics service providers collect information on conditions and driving style. Optimally, these two sources should be put together [25,26].

It is also worth mentioning that driving risk indicators such as speed, breaking, acceleration or RPM, should be accurately measured, otherwise they could be misleading in a prediction system, especially if such a system is designed as a real-time risk warning tool. In our study, we analyzed daily averages and provided summary characterization of the drivers, we expect that measurement error, i.e., accuracy, could have produced wider confidence intervals of our model parameters. A number of weaknesses need to be addressed in this study. For example, we only use OLS and binary logistic regressions, but other linear regression and multiple logistic regression methods should be tried to conduct the regression analyses. In addition, GPS data are not especially well exploited here. In subsequent studies, GPS track data can be usefully employed to vide the driving road grade (urban road or highway) and driving time (day vs. night), so as to analyze driving risks under different driving conditions. Thanks to the detailed processing and analysis of the data, a driving risk event can be defined quickly so as to explore the risks faced. 

## Figures and Tables

**Figure 1 sensors-20-02712-f001:**
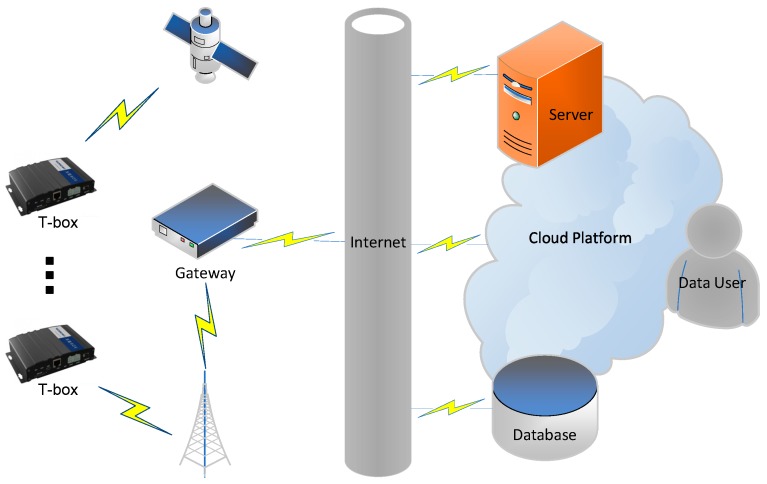
Schematic diagram of data acquisition, transmission, storage and application.

**Figure 2 sensors-20-02712-f002:**
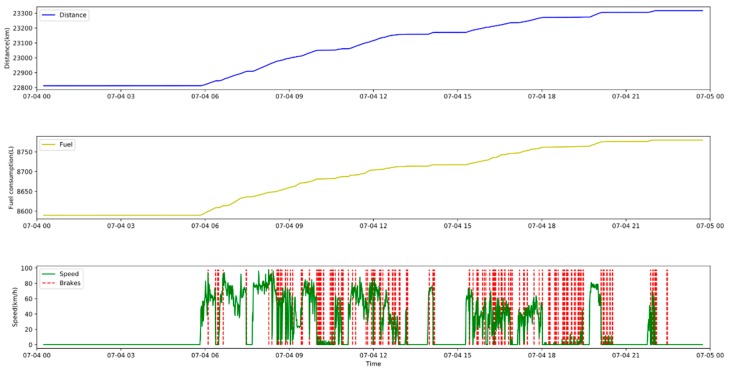
Distance covered, fuel consumption, speed and braking of a vehicle in the course of one day.

**Figure 3 sensors-20-02712-f003:**
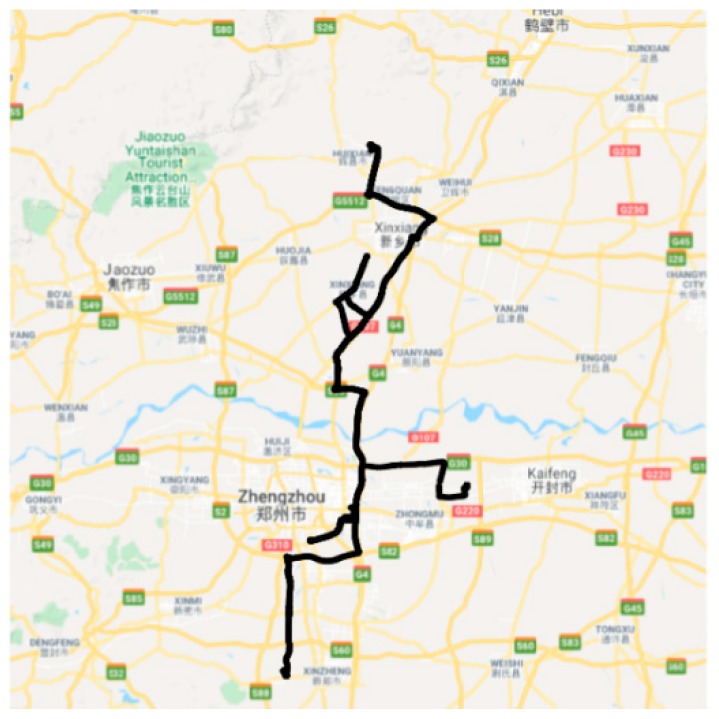
The trajectory of a vehicle in the course of one day.

**Figure 4 sensors-20-02712-f004:**
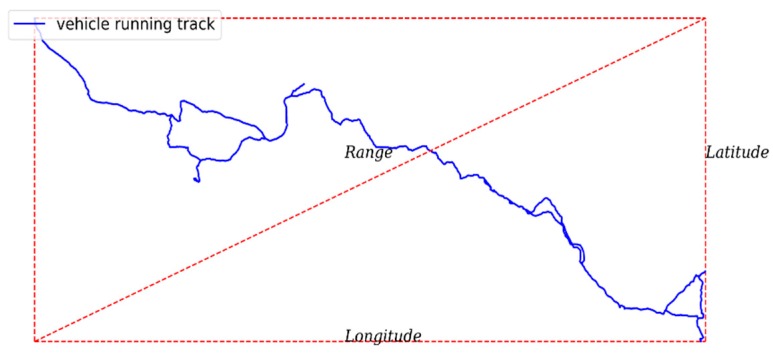
A sample of a trajectory of a driver and the rectangle used to calculate the range.

**Figure 5 sensors-20-02712-f005:**
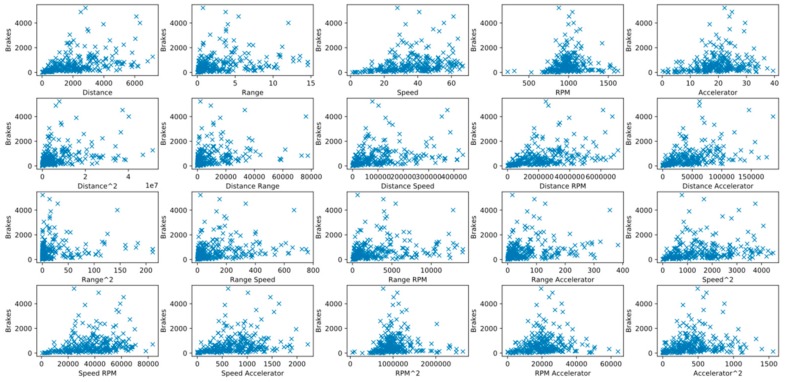
Relationship between “Brakes” and its independent variables.

**Figure 6 sensors-20-02712-f006:**
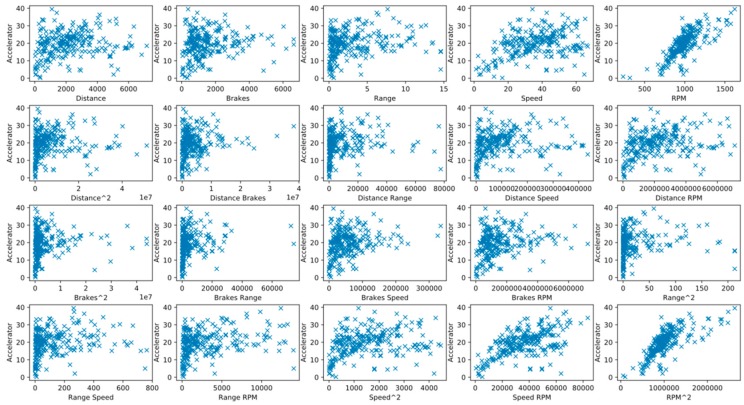
Relationship between “Accelerator” and its independent variables.

**Figure 7 sensors-20-02712-f007:**
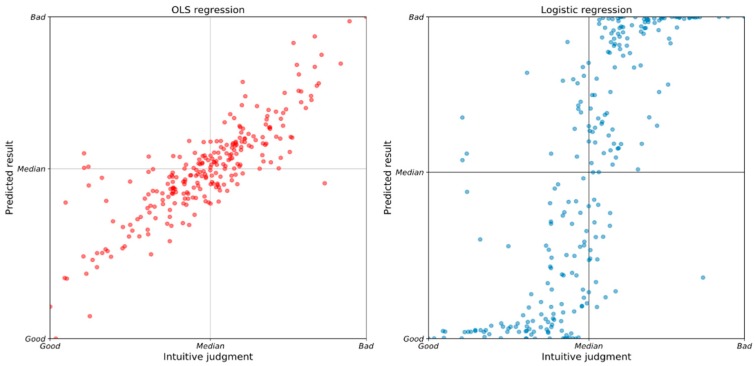
The predicted results were compared with the original data. The horizontal axis presents the original acceleration average level and the vertical axis presents the predicted results in OLS regression (**left**) and logistic regression (**right**) for the whole sample of drivers.

**Figure 8 sensors-20-02712-f008:**
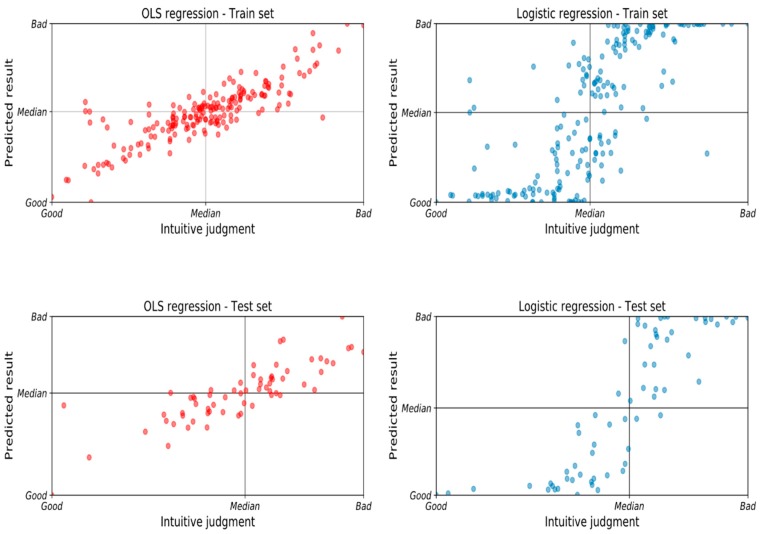
The predicted results were compared with a training (75% in the upper plots) and a test (25% in the lower plots) sample. The horizontal axis presents the original acceleration average level and the vertical axis presents the predicted results in OLS regression (**left**) and logistic regression (**right**).

**Table 1 sensors-20-02712-t001:** Key variables available after data processing.

Key Variables	Definitions
ID	T-box unique identification number
GPS time	GPS recording time
Longitude *	Longitude recorded by GPS
Latitude *	Latitude recorded by GPS
GPS speed	GPS recording speed
RPM	Revolutions per minute recorded by vehicle condition sensor
Accelerator pedal position	Continuous variable, 0(%)—no pedal, 100(%)—pedal to the floor
Brake switch	0/1 variable, 0—no braking, 1—braking
Integral kilometer	Driving distance in the previous 30 s
Integral fuel consumption	Accumulated fuel consumption in the previous 30 s
Interval brake times	Number of brakes in the previous 30 s

* All the records are taken at the instant time and correspond to that second. GPS coordinates ±5 m.

**Table 2 sensors-20-02712-t002:** Summary statistics of the 253 data files ^*^.

	Definition	Mean	Standard Deviation	Minimum	Median	Maximum
Brakes	Brake counts with speed >40 km/h	1540.194	1266.109	26.000	1162.000	6633.000
Accelerator	Mean of acceleration pedal position (%)	19.640	7.313	0.185	20.124	39.480
Distance	Cumulative driving distance (km)	2211.046	1578.700	17.140	1975.570	7163.830
Speed	Mean of speed (km/h)	36.076	15.225	1.187	36.123	66.819
RPM	Mean of revolutions per minute	997.390	178.219	232.263	983.173	1622.257
Range	Range of driving (geographical units)	3.050	3.334	0.013	1.706	14.593

* Each data file produces a vehicle average. This summary describes the averages.

**Table 3 sensors-20-02712-t003:** Regression results with Brakes as a dependent variable.

	OLS Regression		Logistic Regression	
	Coefficient	*p*-Value	Coefficient	*p*-Value
Intercept	−359.5976	0.173	−2.1345	<0.001
Distance	1.4935	0.003	0.0016	<0.001
Speed	−85.1223	0.045		
Accelerator	99.5726	0.014		
Distance^2^	8.574 × 10^−5^	0.012		
Speed^2^			−0.0029	0.001
RPM^2^			−5.309 × 10^−6^	0.005
Accelerator^2^			−0.0094	0.024
Distance × Speed	−0.0180	<0.001		
Distance × RPM	−0.0018	0.006		
Distance × Accelerator	0.0555	<0.001		
Range^2^	−8.7141	0.007		
Range × RPM	0.2533	0.002		
Range × Accelerator	−7.2562	0.018		
Speed × RPM	0.1516	0.003	0.0001	0.045
Speed × Accelerator	−3.1860	0.002		
RPM × Accelerator	−0.0818	0.019	0.0004	0.017
Adjusted-R^2^	0.3420			
Pseudo-R^2^			0.2472	

**Table 4 sensors-20-02712-t004:** Regression results with Accelerator as dependent variable.

	OLS Regression		Logistic Regression	
	Coefficient	*p*-Value	Coefficient	*p*-Value
Intercept	−11.6821	<0.001	−3.6799	<0.001
Distance	−0.9633	<0.001		
Brakes	0.3741	0.009		
Speed	58.8188	<0.001	−46.0734	<0.001
RPM	2.1586	<0.001		
Distance × RPM	0.0885	<0.001		
Brakes × Speed	0.4965	0.002		
Brakes × RPM	−0.0521	0.001		
Speed^2^	−67.6603	<0.001		
Distance^2^			−0.0064	<0.001
Brakes^2^			−0.0025	0.009
Distance × Brakes			0.0085	<0.001
Distance × Range			2.7301	0.001
Brakes × Range			−1.7223	0.003
Range × Speed			−124.8554	0.019
Speed × RPM			6.2241	<0.001
Adjusted-R^2^	0.7070			
Pseudo-R^2^			0.5321	

**Table 5 sensors-20-02712-t005:** Classification results of 253 drivers.

Classification	OLS	Logistic	Both
Low observed value and low predictive score	105	109	101
High observed value and high predictive score	105	109	104
Low observed value and high predictive score	21	17	16
High observed value and low predictive score	22	18	14

**Table 6 sensors-20-02712-t006:** Classification results of training and test data.

Classification	Train Set			Test Set		
	OLS	Logistic	Both	OLS	Logistic	Both
Low observed value and low predictive score	76	78	73	28	30	28
High observed value and high predictive score	76	78	75	28	30	28
Low observed value and high predictive score	18	16	15	4	2	2
High observed value and low predictive score	19	17	14	4	2	2

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
