# Peer review of "Assessing Driving Risk Using Internet of Vehicles Data: An Analysis Based on Generalized Linear Models"

_sensors, 2020, doi:10.3390/s20092712_

Round 1

Reviewer 1 Report

The objective of this simple manuscript is to indicate the relationship between the road traffic accidents and the
indicators of driving risk.In principle, this is an interesting and potentially important study.I have several concerns
regarding the manuscript.

* Are Generalized linear models (GLMs) machine learning (ML) methods?

* Please give the readers a description of the IoV information service provider in China.

* What is the abundant vehicle condition information?

* Please clarify the accuracy of data.

* What is the relationship between the accidents and the indicators of driving risk in your study?

* The authors are not familiar with their study. Please improve the Literature Review part and add some important references.

Author Response

RESPONSE 1.

We thank this reviewer for his helpful comments and suggestions to improve the current version of this manuscript. Here we list the answers to your questions and how we have modified the current version.

Q1. Are Generalized linear models (GLMs) machine learning (ML) methods?

Yes, linear regression and logistic regression are both basic algorithms in machine learning, and they are also included in the category of generalized linear model, which has intersection with machine learning. Therefore, the two generalized linear models we use belong to machine learning. ML methods cover a wide range of classic statistical methods. We apologize for the misunderstanding We have modified the paragraph where this sentence was included and now we say “To date, machine learning (ML) methods, including decision tree, random forest, support vector machine, neural network regressions and generalized linear models (GLMs), among others, have been applied to studies of PHYD plans. However, as most ML algorithms are black-box algorithms, and insurance prices are highly regulated around the world, some ML methods have not gained much popularity. Essentially, black-box procedures cannot be expressed in a predictive parametric formulation so that insurance premiums can be calculated based on an analytic transformation of a weighted combination of factors. GLMs are more familiar to insurance practitioners and customers as regards the practical application of PHYD products. Moreover, GLMs capture the parameterized relationship between driving risk and related factors, which is especially useful for further research into driving risk identification and the communication of features leading to low-risk driving patterns.”

Q2. Please give the readers a description of the IoV information service provider in China.

The IoV information service provider is called ‘China Satellite Navigation and Communications Co., Ltd.’ which is a high-tech enterprise providing intelligent network technology services, headquartered in Beijing. The IoV information service gathers data from their own intelligent network vehicles. This sentence has been inserted in the text (first paragraph of section 3)

Q3. What is the abundant vehicle condition information?

We apologize for having created confusion with the word “abundant”. We have removed this word and we give details about the measurements about the vehicle that are included in the data set (mileage, fuel consumption, speed, acceleration). The sentence now says:

“The data include unique vehicle identification, time-varying GPS trajectory data, and vehicle information (for example, mileage, fuel consumption, speed, accelerator pedal position, brake times and so on); however, they do not include accidents and claims information, which are unavailable for privacy and legal reasons.”

Q4. Please clarify the accuracy of data.

The original data are subject to measurement error. Accuracy in GPS location is +/- 5 meters in general in China, but it some cases it may be +/-30 meters if conditions are not optimal. However, we have aggregated the original data for each vehicle and so, the models are based on averages. We have inserted a note on accuracy in Table 1 when defining the variables.

Q5. What is the relationship between the accidents and the indicators of driving risk in your study?

As mentioned in the literature review, there is an existing body of literature relating accidents with excessive speed, breaking and acceleration. So, even if no accidents have been observed, we assume that there is a significant association between accidents and the indicators of driving risk that we are modelling. We have already cited to articles, mainly published in the journal Accident Analysis and Prevention. We have now included a few more references (see the answer to question Q6 and reviewer 2) and an additional sentence in the conclusions to emphasize the relationship between accidents and the indicators of driving risk. We have inserted the following sentence:

“As mentioned in the literature review, accidents and the indicators of driving risk”in our study are strongly related [-]”.

Q6. The authors are not familiar with their study. Please improve the Literature Review part and add some important references.

We have extended the Literature Review to include a few more references. We would appreciate to receive suggestions from the reviewer and the editor to cover the references that we have not cited. This sentence has been inserted:“Zuo et al. developed a vehicle terminal that can accurately obtain the vehicle’s behavior information, including acceleration, deceleration, sharp turns, and sudden lane change [22]. They suggested that statistical analyses of these data produce a broad application prospect in the field of UBI. Recently, Li et al. proposed a two-factor indicator analysis method to prove that the driver’s physiological information and vehicle dynamic information (especially average speed) have a significant influence on the driving risk [23].”

New references:

Charpentier, A. Computational Actuarial Science with R; 1st ed.; Chapman & Hal: Boca Raton, USA, 2014.

Ayuso, M.; Guillen M.; Nielsen, J.P., Improving automobile insurance ratemaking using telematics: incorporating mileage and driver behaviour data. Transportation 2019, 46, (3), 735-752.

Cheng, B.; Lin, Q.; Song, T.; Cui, Y.; Wang, L.; Kuzumaki, S., Analysis of driver brake operation in near-crash situation using naturalistic driving data. International Journal of Automotive Engineering 2011, 2, 87-94.

Zuo, W.; Guo, C.; Liu, J.; Peng, X.; Yang, M., A Police and Insurance Joint Management System Based on High Precision BDS/GPS Positioning. Sensors 2018, 18, (2), 169.

Li, Y.; Wang, F.; Ke, H.; Wang, L.; Xu, C., A Driver’s Physiology Sensor-Based Driving Risk Prediction Method for Lane-Changing Process Using Hidden Markov Model. Sensors 2019, 19, (12), 2670.

Reviewer 2 Report

p. 2, l. 50: The authors suggest that promoting IoV can improve insurance companies’ profits. This is generally true, but premium differentiation might also trigger selection effects in the portfolio. That’s why there’s also a risk of less profits. A none risk-based increase of premiums could also improve profits. The assumption here is that the total amount of premium underestimated to the total risk. In general, this is not true and where this is not true, another premium differentiation is not a mean of profit increase.

p. 2, l. 75: “much popularity so far”.

p. 3, l. 121/122: “Traditional automobile insurance relies on quite basic information about the customer and the vehicle.”

I wouldn’t agree on that for all insurance markets. The criteria used in tariffs differ from country to country. In Germany the criteria are quite developed and take into account for example vehicle type and regional classes. In addition, mileage is included by estimation of the total amount of kilometers a year (e. g. less than 10.000 km). This is checked, especially when an accident occurs. “Traditional” tariffs might be very close or even better than (just) PAYD tariffs. Moreover, tariffs in motor insurance are often experienced based (with bonus/malus systems depending on claims). That’s why the actual tariff is much depended on the customer (and its driving behavior). A PHYD tariff not based on accident or claims data in general cannot expected to be better in the long run. For short time classification this might be okay, but without experienced based tariffication, I would not expect better results.  

The main driver of better pricing just based on driving behaviour is only accessible if internal data of the vehicle are combined with map/street and weather data. It might be necessary to have also additional data from other sources (other vehicles, traffic jam cameras, speeding limit in areas of road work, etc.).

p. 5, l. 174 – 176: Please provide some indications that weather and other relevant conditions could be assumed as almost identical in these two periods for the two groups.

Author Response

We thank this reviewer for his helpful suggestions that have improved the current version of our manuscript.

We provide an answer to his remarks.

  1. 2, l. 50: The authors suggest that promoting IoV can improve insurance companies’ profits. This is generally true, but premium differentiation might also trigger selection effects in the portfolio. That’s why there’s also a risk of less profits. A none risk-based increase of premiums could also improve profits. The assumption here is that the total amount of premium underestimated to the total risk. In general, this is not true and where this is not true, another premium differentiation is not a mean of profit increase.

We agree with the reviewer that IoV does not always lead to a profit increase. What we see is that insurers can offer new services. We have changed this sentence accordingly.

    p2. l. 75: “much popularity so far”.

                Agree. We have introduced this.

    p3, l. 121/122: “Traditional automobile insurance relies on quite basic information about the customer and the vehicle.” (---reviewers' comments---)

We agree with the reviewer that our discussion was too brief. We have included these comments in our review.

    p5, l. 174 – 176: Please provide some indications that weather and other relevant conditions could be assumed as almost identical in these two periods for the two groups.

Done

Reviewer 3 Report

This paper examines how IoV data can be used to access driver performance. The study uses OLS and binary logistic regression results to discuss driver behaviour, road traffic safety and motor insurance.

The introduction is quite out of step with the paper. The first paragraph discusses the internet of vehicles (IoV). The authors state that the data is collected from multiple sensors and is process and shared and published on the network platform. What network platform and to whom does the vehicle share this data? In the same paragraph you state “...the eIoV effectively guides and supervises vehicles according to different functional requirements and provides professional and personal internet application services”. Are you saying this is an autonomous vehicle? What professional and personal internet services? In the third paragraph you state that “... IoV data are typically collected by means of on-board devices: pre- and post-installed devices and smart phones being the three most common data acquisition terminals”. I assume the installed devices are telematic boxes but they don’t “guide and supervise” the vehicle and “provides professional and personal internet application services”.

“Manage-how-you-drive (MHYD), data processing results and occasional vehicle failure information can be fed back to drivers ensuring they drive safely at all times.” I don’t see how MHYD ensures constant driving safety.

The level of external research citation is poor. For example, you have statements like , “Generalized linear models (GLMs) have been used to score and classify drivers”, “In contrast with most previous studies,...” and “many studies report a positive correlation between …” without any relevant citations.

Figure 2 shows driving behaviour for one 24-hour period. The x-axis is unclear but it looks like the vehicle was in use for 18 hours covering about 900 km. That suggests a commercial vehicle with more than one driver. Fig 3 seems to be a commercial route. If I am correct and there were multiple drivers of the vehicles, then I expect the data gathered to be aggregated to some extent. I think this should be addressed in the text.

The description of harsh braking and acceleration is good but I think some discussion around the speed of the vehicle at the time is important. For example, a harsh braking event can occur when the vehicle is travelling at 5km/hr with no safety implications while the same harsh braking at 50km/hr is obviously a different proposition.

You “observe” two-by-two plots (Figures 5 and 6) and state that it is “apparent that regardless of which dependent variable is chosen, no linear relationship is clear between it and any of the independent variables”. You go on to, eventually, explain the use of step-wise regression. The ordering of your thoughts in this section is somewhat hard to understand.

I would expect some goodness of fit analysis. That is, a section of the data should be test out-of-sample to show how the model predicts new data.

Author Response

We thank this reviewer for his helpful comments and suggestions to improve the current version of this manuscript. Here we list the answers to your questions and how we have modified the current version.

Q1. The introduction is quite out of step with the paper. The first paragraph discusses the internet of vehicles (IoV). The authors state that the data is collected from multiple sensors and is process and shared and published on the network platform. What network platform and to whom does the vehicle share this data? In the same paragraph you state “...the eIoV effectively guides and supervises vehicles according to different functional requirements and provides professional and personal internet application services”. Are you saying this is an autonomous vehicle? What professional and personal internet services? In the third paragraph you state that “... IoV data are typically collected by means of on-board devices: pre- and post-installed devices and smart phones being the three most common data acquisition terminals”. I assume the installed devices are telematic boxes but they don’t “guide and supervise” the vehicle and “provides professional and personal internet application services”.

We apologize to have created such a confusion. We have now explained that he network platform belongs to insurance entities. This is not autonomous vehicles. We have removed the provision of professional and personal internet applications (this referred to mobile apps and remote control of commercial vehicles). We do not think that this is relevant to our study. We thank the reviewer for help us improve the introduction we have removed the “guide and supervise” concept and the “provides professional and personal internet application services”. We now simply state that the installed devices are thematic boxes and we emphasize that these are not autonomous vehicles.

Q2 “Manage-how-you-drive (MHYD), data processing results and occasional vehicle failure information can be fed back to drivers ensuring they drive safely at all times.” I don’t see how MHYD ensures constant driving safety.

We agree that the sentence is confusing. MHYD only provides summary information like average speed, total number of hours driven. This is informative and it does not “per se” ensure safety.

Q3 The level of external research citation is poor. For example, you have statements like , “Generalized linear models (GLMs) have been used to score and classify drivers”, “In contrast with most previous studies,...” and “many studies report a positive correlation between …” without any relevant citations.

We have increased the level of external research citations. Now the sentences mentioned by the referee, have the corresponding citations:

“To date, machine learning (ML) methods, including decision tree, random forest, support vector machine, neural network regressions and generalized linear models (GLMs), among others, have been applied to studies of PHYD plans [].”

“In contrast with most previous studies [13,20], the driving behavior data used here to identify risky drivers are short-term and do not include accidents and claims.”

“For example, many studies report a positive correlation between accidents and excessive acceleration and braking and so, here, we use these data to undertake our analysis of driving risk [].”

And the following references have been added to the article.

Charpentier, A. Computational Actuarial Science with R; 1st ed.; Chapman & Hal: Boca Raton, USA, 2014.

Ayuso, M.; Guillen M.; Nielsen, J.P., Improving automobile insurance ratemaking using telematics: incorporating mileage and driver behaviour data. Transportation 2019, 46, (3), 735-752.

Cheng, B.; Lin, Q.; Song, T.; Cui, Y.; Wang, L.; Kuzumaki, S., Analysis of driver brake operation in near-crash situation using naturalistic driving data. International Journal of Automotive Engineering 2011, 2, 87-94.

Q4 Figure 2 shows driving behaviour for one 24-hour period. The x-axis is unclear but it looks like the vehicle was in use for 18 hours covering about 900 km. That suggests a commercial vehicle with more than one driver. Fig 3 seems to be a commercial route. If I am correct and there were multiple drivers of the vehicles, then I expect the data gathered to be aggregated to some extent. I think this should be addressed in the text.

We thank the reviewer for pointing out that the text was incorrect. Even if the vehicle was monitored all day, it is clear in Figure 2 (bottom plot) that the car is idle most of the time (when speed is equal to zero). The car is used early in the morning and in the afternoon. We have extended this explanation in the text. The sentence now says:

Figure 2 shows the operation of a vehicle from 0:00 on July 4, 2018 to 0:00 on July 5, 2018. The vehicle’s recorded mileage increased from 22,812.35 km to 23,317.339 km, and its accumulated consumption increased from 8,589.69 L to 8,779.67 L. At the lower part of Figure 2, we display the speed. We see that the car is idle during the night.

Q5 The description of harsh braking and acceleration is good but I think some discussion around the speed of the vehicle at the time is important. For example, a harsh braking event can occur when the vehicle is travelling at 5km/hr with no safety implications while the same harsh braking at 50km/hr is obviously a different proposition.

We agree with the reviewer that this discussion is important and this is the reason why we need to look at the detailed data. With one summary per driver, we only analyse averages during the day, but not the interaction between these variables n real time. The following sentence has been inserted in the conclusions.

“Some discussion around the speed of the vehicle at the time is important. For example, a harsh braking event can occur when the vehicle is travelling at 5km/hr with no safety implications while the same harsh braking at 50km/hr is obviously a different proposition. We need to analyse the detailed data instead of averages.”

Q6 You “observe” two-by-two plots (Figures 5 and 6) and state that it is “apparent that regardless of which dependent variable is chosen, no linear relationship is clear between it and any of the independent variables”. You go on to, eventually, explain the use of step-wise regression. The ordering of your thoughts in this section is somewhat hard to understand.

We agree with the reviewer that this is confusing. We have removed the sentence in the new version.

Q7. I would expect some goodness of fit analysis. That is, a section of the data should be test out-of-sample to show how the model predicts new data.

We thank the reviewer for this suggestions. We have now taken 75% drivers randomly and we have tested the models with 25% test sample. The conclusions do not vary. We report Figure 8 and Table 6 using train set and test set. A sentence mentioning this analysis is included in the discussion.

New material: Figure 7. and Table 6 inserted

Classification

Train set

Test set

OLS

Logistic

Both

OLS

Logistic

Both

Low observed value and low predictive score

76

78

73

28

30

28

High observed value and high predictive score

76

78

75

28

30

28

Low observed value and high predictive score

18

16

15

4

2

2

High observed value and low predictive score

19

17

14

4

2

2

Round 2

Reviewer 1 Report

I appreciate the time the authors have taken to address my concerns. In principle, the project is an interesting and potentially important study. Unfortunately, I continue to have several concerns regarding the study.

* What is the relationship between the accidents and the indicators of driving risk in your study? And does the accuracy of data affect the relationship between the accidents and the indicators of driving risk in your study?

* The language of the paper needs further improvement. For instance, “these features” should be explained (line 32); there exist grammatical problems in the sentence “it is critical that driving risk be accurately and effectively assessed” (line 63); the “latmax” appears twice in Equation (1); etc.

Author Response

We have addressed the two questions as follows:

* What is the relationship between the accidents and the indicators of driving risk in your study? And does the accuracy of data affect the relationship between the accidents and the indicators of driving risk in your study?

We have included some additional references that provide evidence on the relationship between accidents and the indicators of driving risk in our study (speeding, acceleration/braking). Some references were suggested by Reviewer 3. We have also added to our discussion the following sentence:

To analyze telematics data and accident data, a comprehensive view of risky driving is needed, Insurers tend to have a complete vision of accidents and in particular, accident severity combined with the driver’s past experience. Telematics service providers collect information on conditions and driving style. Optimally, these two sources should be put together.

Regarding the accuracy of data and relationship between the accidents and the indicators of driving risk, we have included the following comment:

It is also worth mentioning that driving risk indicators such as speed, breaking, acceleration or RPM, should be accurately measured, otherwise they could be misleading in a prediction system, especially if such a system is designed as a real-time risk warning tool. In our study we analyzed daily averages and provided summary characterization of the drivers, we expect that measurement error, ie. accuracy, could have produced wider confidence intervals of our model parameters.

New references:

Ryan, C., Murphy, F. and Mullins, M., 2020. Spatial risk modelling of behavioural hotspots: Risk-aware path planning for autonomous vehicles. Transportation Research Part A: Policy and Practice, 134, pp.152-163.

Ayuso, M., Guillen, M. and Marín, A.M.P., 2016. Using GPS data to analyse the distance travelled to the first accident at fault in pay-as-you-drive insurance. Transportation research part C: emerging technologies, 68, pp.160-167.

Boucher, J.P., Côté, S. and Guillen, M., 2017. Exposure as duration and distance in telematics motor insurance using generalized additive models. Risks, 5(4), p.54.

* The language of the paper needs further improvement. For instance, “these features” should be explained (line 32); there exist grammatical problems in the sentence “it is critical that driving risk be accurately and effectively assessed” (line 63); the “latmax” appears twice in Equation (1); etc.

Our English correction service checked the grammatical correctness of this version. We have also corrected the sentence in line 32. We gave also corrected line 63 and we have checked Equation (1). We thank the reviewer for heling us improve this manuscript.

Reviewer 3 Report

The text is much improved. I have some remaining issues that must be addressed.

1) In the text you state "During this period, driving speeds varied from 0
to 110 km per hour" however in Table 2 the maximum speed observed is 66.819. Can you explain the discrepancy?

2) The average number of engine RPMs is 997.39 with a maximum of 1622. Those figures are very low. Can this be explained?

3) Please increase the caption description in figure 7 and figure 8

4) The inclusion of an out-of-sample data set is welcome. Please explain how the data was split

5) I suggested some useful references in my first report and these were not used.

Verbelen, R., Antonio, K. and Claeskens, G., 2018. Unravelling the
predictive power of telematics data in car insurance pricing. Journal of
the Royal Statistical Society: Series C (Applied Statistics), 67(5),
pp.1275-1304.

Ryan, C., Murphy, F. and Mullins, M., 2020. Spatial risk modelling of
behavioural hotspots: Risk-aware path planning for autonomous vehicles.
Transportation Research Part A: Policy and Practice, 134, pp.152-163.

Ayuso, M., Guillen, M. and Marín, A.M.P., 2016. Using GPS data to
analyse the distance travelled to the first accident at fault in
pay-as-you-drive insurance. Transportation research part C: emerging
technologies, 68, pp.160-167.

Boucher, J.P., Côté, S. and Guillen, M., 2017. Exposure as duration and
distance in telematics motor insurance using generalized additive
models. Risks, 5(4), p.54.

Author Response

We have addressed the questions as follows:

  • In the text you state "During this period, driving speeds varied from 0 to 110 km per hour" however in Table 2 the maximum speed observed is 66.819. Can you explain the discrepancy?

Indeed, Table 2 represents the daily average. So, maximum equal to 66.819 in Table 2 refers to a maximum of daily averages not to the maximum speed observed at a certain point during the day. We inserted a not at the bottom of Table 2 “This summary describes the averages.”

  • The average number of engine RPMs is 997.39 with a maximum of 1622. Those figures are very low. Can this be explained?

Again, we believe these low averages are obtained as a result of averaging over the entire day so that when the car is idle, RPMs I actually equal to 0, so the resulting average is low. We have inserted this comment in the text.

RPM is normal between 0 and 600 when the vehicle is idling, and 0 when the vehicle is just stalling, which reduces the average RPM to a certain extent. In the following studies, we will consider eliminating these 0 values, but the impact on the mean value will not be too great, and this will not affect the model establishment and analysis of this study.

  • Please increase the caption description in figure 7 and figure 8

We inserted the following caption in Figure 7. The horizontal axis presents the original Acceleration average level and the vertical axis presents the predicted results in OLS regression (left) and logistic regression (right) for the whole sample of drivers.

We inserted the following caption in Figure 8. The predicted results were compared with a training (75% in the upper plots) and a test (25% in the lower plots) sample. The horizontal axis presents the original Acceleration average level and the vertical axis presents the predicted results in OLS regression (left) and logistic regression (right).

  • The inclusion of an out-of-sample data set is welcome. Please explain how the data was split.

We have introduced the following explanation.

A section of the data should be test out-of-sample to show how the model predicts new data. We have split the sample in a train data set (75% instances) and a test data set (25% instances). The train and test samples were chosen randomly.

  • I suggested some useful references in my first report and these were not used.

We apologize for this problem. We actually had not received these references. We have requested them to the editor and the following list was provided. Note that reference (a) was already included. We have inserted (b), (c) and (d).

(a) Verbelen, R., Antonio, K. and Claeskens, G., 2018. Unravelling the predictive power of telematics data in car insurance pricing. Journal of the Royal Statistical Society: Series C (Applied Statistics), 67(5), pp.1275-1304.

(b) Ryan, C., Murphy, F. and Mullins, M., 2020. Spatial risk modelling of behavioural hotspots: Risk-aware path planning for autonomous vehicles. Transportation Research Part A: Policy and Practice, 134, pp.152-163.

(c) Ayuso, M., Guillen, M. and Marín, A.M.P., 2016. Using GPS data to analyse the distance travelled to the first accident at fault in pay-as-you-drive insurance. Transportation research part C: emerging technologies, 68, pp.160-167.

(d) Boucher, J.P., Côté, S. and Guillen, M., 2017. Exposure as duration and distance in telematics motor insurance using generalized additive models. Risks, 5(4), p.54.